# Metabolomic and Mitochondrial Fingerprinting of the Epithelial-to-Mesenchymal Transition (EMT) in Non-Tumorigenic and Tumorigenic Human Breast Cells

**DOI:** 10.3390/cancers14246214

**Published:** 2022-12-16

**Authors:** Elisabet Cuyàs, Salvador Fernández-Arroyo, Sara Verdura, Ruth Lupu, Jorge Joven, Javier A. Menendez

**Affiliations:** 1Metabolism and Cancer Group, Program Against Cancer Therapeutic Resistance (ProCURE), Catalan Institute of Oncology, 17005 Girona, Spain; 2Girona Biomedical Research Institute, 17190 Girona, Spain; 3Eurecat, Centre Tecnològic de Catalunya, Centre for Omic Sciences, Joint Unit Eurecat-Universitat Rovira I Virgili, Unique Scientific and Technical Infrastructure (ICTS), 43204 Reus, Spain; 4Department of Laboratory Medicine and Pathology, Division of Experimental Pathology, Mayo Clinic, Rochester, MN 55905, USA; 5Department of Biochemistry and Molecular Biology Laboratory, Mayo Clinic Minnesota, Rochester, MN 55905, USA; 6Mayo Clinic Cancer Center, Rochester, MN 55905, USA; 7Unitat de Recerca Biomèdica (URB-CRB), Hospital Universitari de Sant Joan, Institut d’Investigació Sanitaria Pere Virgili, Universitat Rovira I Virgili, 43204 Reus, Spain

**Keywords:** metabolism, mitochondria, phenotypic screening, breast cancer, therapy resistance, complex III

## Abstract

**Simple Summary:**

Epithelial-to-mesenchymal transition (EMT) is a cellular program that enables epithelial cells to transition toward a mesenchymal phenotype with augmented cellular motility. Although EMT is a fundamental, non-pathological process in embryonic development and tissue repair, it also confers biological aggressiveness to cancer cells, including invasive behavior, tumor- and metastasis-initiating cancer stem cell activity, and greater resistance to all the cancer treatment modalities. Whereas alterations in the metabolic microenvironment are known to induce EMT, it is also true that the EMT process involves a very marked metabolic remodeling. However, whether there is a causal or merely an ancillary relationship between metabolic rewiring and the EMT phenomenon has not yet been definitively clarified. Here, we combined several technology platforms to assess whether the accompanying changes in the metabolic profile and mitochondria functioning that take place during the EMT process are independent or not of the non-tumorigenic versus tumorigenic nature of epithelial cells suffering a mesenchymal conversion. Understanding the metabolic basis of the non-tumorigenic and tumorigenic EMT provides fundamental insights into the causation and progression of cancer and may, in the long run, lead to new therapeutic strategies.

**Abstract:**

Epithelial-to-mesenchymal transition (EMT) is key to tumor aggressiveness, therapy resistance, and immune escape in breast cancer. Because metabolic traits might be involved along the EMT continuum, we investigated whether human breast epithelial cells engineered to stably acquire a mesenchymal phenotype in non-tumorigenic and H-Ras^V12^-driven tumorigenic backgrounds possess unique metabolic fingerprints. We profiled mitochondrial–cytosolic bioenergetic and one-carbon (1C) metabolites by metabolomic analysis, and then questioned the utilization of different mitochondrial substrates by EMT mitochondria and their sensitivity to mitochondria-centered inhibitors. “Upper” and “lower” glycolysis were the preferred glucose fluxes activated by EMT in non-tumorigenic and tumorigenic backgrounds, respectively. EMT in non-tumorigenic and tumorigenic backgrounds could be distinguished by the differential contribution of the homocysteine-methionine 1C cycle to the transsulfuration pathway. Both non-tumorigenic and tumorigenic EMT-activated cells showed elevated mitochondrial utilization of glycolysis end-products such as lactic acid, β-oxidation substrates including palmitoyl–carnitine, and tricarboxylic acid pathway substrates such as succinic acid. Notably, mitochondria in tumorigenic EMT cells distinctively exhibited a significant alteration in the electron flow intensity from succinate to mitochondrial complex III as they were highly refractory to the inhibitory effects of antimycin A and myxothiazol. Our results show that the bioenergetic/1C metabolic signature, the utilization rates of preferred mitochondrial substrates, and sensitivity to mitochondrial drugs significantly differs upon execution of EMT in non-tumorigenic and tumorigenic backgrounds, which could help to resolve the relationship between EMT, malignancy, and therapeutic resistance in breast cancer.

## 1. Introduction

Epithelial-to-mesenchymal transition (EMT), a developmental program that primes cells for subsequent cell fate conversion, was proposed as a key requirement for invasion and metastasis to distant organs more than 10 years ago [1,2,3,4,5,6,7]. This conceptual framework has been challenged by fate-mapping studies suggesting that EMT might be dispensable for metastatic outgrowth [8,9,10]. Indeed, cancer cells can metastasize via stable hybrid E/M phenotypes possessing higher stem-like tumor initiation properties and metastatic potential as compared to cells on either end of the EMT spectrum [11,12,13,14]. However, the evidence supporting a role for EMT in conferring therapeutic resistance is clear and compelling [15,16,17,18,19,20,21]. The correlation of lower survival with an activated EMT program in patients with residual disease is now viewed as the consequence of the capacity of EMT cells to resist a broad spectrum of therapeutic interventions such as hormonal therapy, chemotherapy, radiotherapy, and many targeted therapies including immunotherapy [22,23,24,25,26,27]. Accordingly, the successful therapeutic manipulation of EMT has tremendous clinical potential, and different approaches have been proposed to reprogram EMT, including the unlock of fully differentiated mesenchymal states, promotion of (re)epithelial differentiation, and targeting of EMT markers [24,25,26,27]. These strategies might involve inhibition of both cell-autonomous EMT drivers (e.g., growth factor signaling, epigenetic reprogramming, transcription factors, and microRNAs) and non-cell autonomous EMT-driving factors such as non-malignant stromal cells and non-cellular elements [24,25,26,27]. However, pharmacological inhibition of EMT-associated effectors at the molecular level remains challenging, and none of the aforementioned strategies have translated into approved therapies.

A growing number of studies have provided examples of how specific metabolic traits might represent integral parts of the EMT program [28,29,30,31,32,33,34,35,36,37,38,39,40]. A causal relationship between metabolic rewiring and EMT induction has been mostly explored in cancers with genetic deficiencies in metabolic enzymes such as fumarate hydratase (FH), isocitrate dehydrogenase (IDH), and succinate dehydrogenase (SDH) [30,32,34]. Genetic approaches to identifying the metabolic determinants of the EMT process have revealed the requirement of reprogrammed gluconeogenesis (via suppression of the gluconeogenesis rate-limiting enzyme fructose-1,6-bisphosphatase [41] and the key glycolytic enzyme phosphofructokinase-1 [42]) and nucleotide (via promotion of the pyrimidine-degrading enzyme dihydropyridine dehydrogenase [43]) pathways to support the mesenchymal phenotype. We previously reported that EMT-activated cells acquire the ability to metabolize high-energy nutrients such as glycolysis end-products and ketone bodies to support mitochondrial energy production [29]. Given that mitochondria are increasingly recognized as key signaling regulators of cell fate in physiological and pathological conditions [44], it is reasonable to suggest that shifts in mitochondrial function and dynamics might mechanistically resolve the causative versus bystander nature of the established correlation between metabolic reprogramming and EMT [45,46,47,48,49,50]. Moreover, metabolic targeting of EMT might be possible if the vulnerabilities in the distinct utilization of cellular metabolites between the epithelial and mesenchymal cellular state can be identified. Indeed, an underexplored possibility is that the mechanistic coupling between metabolism, (re)programming, and EMT also involves resistance to metabolic poisons with well-characterized mechanisms of action.

Here, we aimed to provide a comprehensive metabolomic and mitochondrial fingerprinting of the EMT phenomenon in human breast cells. We performed targeted metabolomics characterization coupled to functional mitochondrial phenotyping of the human breast EMT in non-tumorigenic and oncogenic H-Ras^V12^-driven tumorigenic backgrounds (Figure 1). One phenotypic dimension was based on targeted analyses of metabolites representative of the catabolic/anabolic status of mitochondrial nodes and of the methionine/folate one-carbon (1C) cycle. A second phenotypic dimension was based on mitochondria-focused functional assays (MitoPlate™) that measures the rate of electron flow into and through the electron transport chain (ETC) from 31 different mitochondrial substrates that produce NAD(P)H or FADH_2_ as well as the mitochondrial sensitivity to a panel of 22 diverse mitochondria-centered inhibitors. We provide evidence that non-tumorigenic and tumorigenic EMT cell fate conversion is accompanied by a conspicuous but distinctive rewiring of metabolic and mitochondria functioning.

## 2. Materials and Methods

### 2.1. Cell Lines and Culture

HMLE^shControl^, HMLE^shECad^, HMLER^shCntrol^, and HMLER^shEcad^ cells were gifts from Prof. Robert A. Weinberg (Whitehead Institute for Biomedical Research, Cambridge, MA). Cells were cultured in a 1:1 mixture of MEGM (Mammary Epithelial Bullet Kit, ref. H3CC-3150; Lonza, Basel, Switzerland) and DMEM/F12 supplemented with 10% fetal bovine serum (FBS), insulin (10 μg/mL), hydrocortisone (0.5 μg/mL), hEGF (10 ng/mL), 1% L-glutamine, and penicillin/streptomycin (Sigma, Madrid, Spain). All cells were tested for mycoplasma contamination using a PCR-based assay prior to experimentation and were intermittently tested thereafter.

### 2.2. Mammosphere Culture and Mammosphere-Forming Efficiency (MSFE)

For mammosphere formation, single cell suspensions of HMLER^shCntrol^ and HMLER^shEcad^ cells were seeded at 1000 cells/cm^2^ in six-well ultralow attachment plates (Corning Inc., New York, NY, USA). Mammosphere medium consisted of serum-free F12/DMEM containing 5 mg/mL insulin, 0.5 mg/mL hydrocortisone, 2% B27 supplement (Invitrogen Ltd., Carslbad, CA, USA), and 20 ng/mL epidermal growth factor (Sigma, Madrid, Spain). The medium was made semi-solid by the addition of 0.5% methylcellulose (R&D Systems, Minneapolis, MN, USA) to prevent cell aggregation. The tumorsphere-forming efficiency (TFE) was calculated after 7 days using the following equation:$$\mathrm{TFE}\left(\%\right)=\frac{\#\;\mathrm{of}\;\mathrm{tumorspheres}\;(\mathrm{large}\;\mathrm{diameter}\;>50\;\mu m)\;\mathrm{per}\;\mathrm{well}}{\#\;\mathrm{of}\;\mathrm{cells}\;\mathrm{seeded}\;\mathrm{per}\;\mathrm{well}}\;\times \;100$$

### 2.3. Flow Cytometry

Cells were washed once with phosphate-buffered saline (PBS) and then harvested with 0.05% trypsin/0.025% EDTA into single cell suspensions. Detached cells were washed with PBS containing 1% FBS and 1% penicillin/streptomycin (wash buffer), counted and resuspended in the wash buffer (10^6^ cells/100 μL). Combinations of fluorochrome-conjugated monoclonal antibodies obtained from BD Pharmingen against human CD44 (PerCP-Cy™ 5.5, Mouse anti-human, ref. No. 560531) and CD24 (PE Mouse Anti-Human, ref. No. 555428) or their respective isotype controls (PerCP-Cy™ 5.5 Mouse IgG2b, κ Isotype Control, ref. No. 558304; PE Mouse IgG2a, κ Isotype Control, ref. No. 556653) were added to the cell suspensions at concentrations recommended by the manufacturer and incubated at 4 °C in the dark for 30–40 min. Labeled cells were washed in the wash buffer to eliminate unbound antibodies, then fixed in PBS containing 1% paraformaldehyde, and then analyzed no longer than 1 h post-staining on a Becton Dickinson Accuri C6 flow cytometer. Data were analyzed using Accuri C6 Flow software.

### 2.4. Targeted Metabolomics and Data Analysis

For targeted metabolomic experiments, HMLE^shControl^, HMLE^shECad^, HMLER^shCntrol^, and HMLER^shEcad^ cells were plated in 6-well plates with normal growth medium, which was replaced after 18 h with complete fresh medium; cells were then incubated under standard cell culture conditions for additional 48 h (*n* = 5 biological replicates in triplicate). Quantitative measurement of up to 30 selected bioenergetic metabolites representative of the catabolic and anabolic status of mitochondria-related metabolic nodes was performed by employing a previously described GC-EI-QTOF-MS method [51,52,53,54]. Quantitative measurement of up to 14 selected metabolites representative of the methionine/folate bi-cyclic 1C metabolome was performed by employing a previously described UHPLC-ESI-QqQ-MS/MS method [55,56,57]. To measure energy metabolism-related metabolites, cell pellets were resuspended in 200 μL methanol/water (8:2) and D4-succinic acid and then lysed with three cycles of freezing and thawing using liquid N_2_ and sonicated with three cycles of 30 s. Samples were maintained in ice for 1 min between each sonication step. Proteins were precipitated, samples were centrifuged, and supernatant was collected. After metabolite extraction, samples were dried under N_2_ and derivatized to rapidly form silyl derivatives using methoxyamine hydrochloride dissolved in pyridine (40 mg/mL) and N-methyl-N-trimethylsilyl trifluoroacetamide. We used a 7890A gas chromatograph coupled with an electron impact source to a 7200 quadrupole time-of-flight mass spectrometer (Agilent Technologies, Santa Clara, CA, USA). For measuring 1C metabolites, extraction was carried out by resuspending the cell pellets in 200 μL of methanol/water (8:2) containing 1% ascorbic acid (*m*/*v*) and 0.5% β-mercaptoethanol (*v*/*v*). Cells were lysed using the same lysis procedure described above; after protein precipitation, samples were centrifuged, and the supernatants were dried under N_2_ and then resuspended in ultrapure water containing 50 mmol/L ammonium acetate and 0.2% formic acid. The analysis was performed with an ultra-high pressure liquid chromatography-quadrupole time-of-light mass spectrometer (Agilent Technologiesm, Santa Clara, CA, USA). Raw data were processed, and compounds were detected and quantified using the Qualitative and Quantitative Analysis B.06.00 software (Agilent Technologies), respectively. Multivariate analysis was applied to pattern recognition, including supervised PLS-DA. The relative magnitude of observed changes was evaluated using VIP scores. Statistical significance was set at *p*  ≤  0.05. MetaboAnalyst 4.0 program (available on the web: http://www.metaboanalyst.ca/ accessed on 1 June 2022) was used to generate scores/loading plots, heatmaps, and multivariate random forest analyses [58].

### 2.5. Mitochondrial Function Phenotyping

Mitochondrial activity was measured in triplicate using 96-well MitoPlate™ S-1 plates (Cat. #14105, Biolog, Hayward, CA, USA). Wells containing the different cytoplasmic and mitochondrial metabolic substrates (*n* = 31) were rehydrated with a solution containing mitochondrial assay solution (MAS) (Biolog cat. #72303), redox dye MC (Biolog cat. # 74353), and 30 μg/mL saponin (Sigma, cat. #84510) in sterile water. Cells were washed with PBS and resuspended in 1× Biolog MAS and added to each well at a final cell density of 30,000 cells/well. Metabolism of substrates was assessed by monitoring colorimetric change of the terminal electron acceptor tetrazolium redox dye at a wavelength of 590 nm on a kinetic microplate reader (2 h).

### 2.6. Mitochondrial Drug Phenotyping

Responsiveness to mitochondria-centered drugs was measured using 96-well MitoPlate™ I-1 plates (Cat. #14104, Biolog, Hayward, CA, USA) following the manufacturer’s instructions. Briefly, wells containing the different mitochondrial inhibitors were rehydrated with a solution containing redox dye MC, 30 μg/mL saponin, and 96 mmol/L succinate (Sigma, cat. #S2378) for 1 h at 37 °C. Cells were washed with PBS (1×) and resuspended at a density of 10^5^ cells/30 μL using 1× Biolog MAS and added to each well. The MitoPlate™ I-1 plate was then loaded on a microplate reader for kinetic reading every 2 h. Alternatively, we omitted the saponification step and cells were cultured for 48 h in white DMEM medium before assessing cell viability by monitoring colorimetric changes at 590 nm for 2 h.

### 2.7. Statistical Analysis

Results from targeted metabolomics were compared by one-way ANOVA with Dunnett’s multiple pair-wise comparison tests using a significance threshold of 0.05. Other calculations including comparisons with the Mann–Whitney U test and/or correlations were made using GraphPad Prism software 6.01 (GraphPad Software, San Diego, CA, USA).

## 3. Results

To avoid the confounding effects of significant differences in genetic backgrounds when employing non-EMT versus EMT-like cancer cell lines, or the possibility of generating “artificial phenotypes” by forced overexpression of EMT-driving transcription factors, we took advantage of two well-characterized models of EMT generated from mammary epithelial cells [59]. The experimental system is based on primary human mammary epithelial cells (HMECs) with sequential retroviral-mediated expression of the telomerase catalytic subunit (generating HMEC/hTERT cells), SV40 large T and small t antigens (generating HMLE cells), and the oncogenic H-Ras allele H-Ras^V12^ (generating HMLER cells). Non-tumorigenic HMLE and tumorigenic HMLER cells were modified by short hairpin RNA-mediated inhibition of *CDH1* encoding E-cadherin, triggering EMT and resulting in the stable acquisition of a mesenchymal phenotype with significantly increased drug resistance [59,60]. Before using the pairs of HMLE^shControl^/HMLE^shECad^ (hereinafter named HMLE/HMLE-EMT) and HMLER^shControl^/HMLER^shECad^ (hereinafter named HMLER/HMLER-EMT) for targeted metabolomic characterization and functional mitochondrial phenotyping of the breast EMT program in non-tumorigenic and tumorigenic backgrounds, we aimed to confirm the presence of their originally described EMT-like phenotypic traits [59,60,61] (Figure 2).

When expanded in adherent conditions, phase contrast images confirmed that HMLE and HMLER cells grew in monolayer culture as tightly packed epithelial clusters with typical cobblestone morphology. Knock-down of E-cadherin in HMLE-EMT cells resulted in a more elongated shape and cell scattering in cell subpopulations that acquired a spindle-like morphology (Figure 2A). Fluorescence-activated cell sorting (FACS) using CD44 and CD24 as markers revealed that HMLE-EMT cultures likewise contained a distinct subpopulation of cells carrying the CD44^+^CD24^low/−^ antigenic phenotype associated with so-called mesenchymal cancer stem cells (CSC) [62,63]. An extreme elongated fibroblast-like morphology and almost complete loss of cell–cell contacts was observed in a majority of HMLER-EMT cells. This gaining of a mesenchymally transdifferentiated phenotype was accompanied by notorious acquisition of the CD44^+^CD24^low/−^ antigenic phenotype (Figure 2A). Because an increase in the proportion of HMLE cells that display a mesenchymal morphology has been reported to occur upon serial passaging (>8 weeks) [64], all the metabolomic/mitochondrial characterizations were carried out with HMLE/HMLE-EMT and HMLER/HMLER-EMT pairs cultured for less than 6 weeks to prevent spontaneous conversion of epithelial to mesenchymal cells during prolonged culture. The tumorigenicity of HMLER cells was originally confirmed by the Weinberg group following subcutaneous or orthotopic injection into the mammary glands of immunocompromised mice [61]. To verify an enhanced tumorigenic behavior of HMLER-EMT cells, we compared the capabilities of HMLER and HMLER-EMT cells to form multicellular “microtumors” in non-adherent and non-differentiations conditions (i.e., tumorspheres), a property associated with the presence of mammary stem/progenitor cells with tumor-initiating capacity [65,66]. HMLER-EMT cells showed a highly-significant increase (>30-fold) in the tumorsphere-forming capacity relative to HMLER parental cells (Figure 2B).

### 3.1. Carbon Metabolites in the Upper and Lower Chains of the Glycolytic–Gluconeogenic Reaction Pathway Are Differentially Affected by EMT in Non-Tumorigenic and Tumorigenic Background

We utilized an in-house targeted metabolomics platform coupling gas chromatography to quadrupole time-of-flight mass spectrometry and an electron impact source (GC-EI-QTOF-MS) to simultaneously measure up to 30 selected metabolites representative of the catabolic and anabolic status of mitochondria-related metabolic nodes [51,52,53,54]. The metabolites included representatives of glycolysis and the mitochondrial tricarboxylic acid (TCA) cycle, in addition to other biosynthetic routes such as the pentose phosphate pathway, amino acid metabolism, and de novo fatty acid biogenesis.

We performed a quantitative, comparative assessment of metabolites in HMLE-EMT and HMLER-EMT cells and in HMLE and HMLER parental counterparts (Appendix A). A schematic view of the mean fold-change is presented in Figure 3 (left panels). Analysis of significant metabolic changes occurring post-EMT revealed that the levels of glucose-6-phosphate and fructose 1,6-bisphosphate were significantly higher in HMLE-EMT cells than in HMLE controls. Likewise, we found a significant increase in phosphoenolpyruvate accompanied by decreases in fructose 1,6-bisphosphate, succinate, and citrate in HMLER-EMT cells as compared with HMLER control cells (Appendix A; Figure 3, left panels).

To better analyze our findings, we embedded the metabolic data from pre-/post-EMT pairs into a partial least squares discriminant analysis (PLS-DA) model to use the power of metabolite abundance in group discrimination and prevent type 2 statistical errors when analyzing data at the specific metabolite level. Metabolite-based clustering obtained by PLS-DA using two-dimensional score plots revealed a clear, non-overlapping differentiation of both HMLE-EMT and HMLER-EMT cells from HMLE and HMLER controls (Figure 3, middle panels). To identify the metabolites with the most relevant changes post-EMT, we calculated the variable importance of projection (VIP) scores as a measure of the variable’s degree-of-alteration associated with the acquired EMT status: a higher VIP score was considered more relevant in mesenchymal versus epithelial status classification. When VIP scores ≥ 1.5 in the PLD-DA model were chosen to maximize the difference of metabolic profiles between post-EMT and epithelial parental counterparts, all “6-carbon” metabolites connecting glucose to glyceraldehyde-3-phosphate (G3P) in the upper chain of the glycolytic–gluconeogenic reaction pathway (i.e., glucose-6-phosphate, fructose-6-phosphate, and fructose 1,6-bisphosphate) showed the most relevant post-EMT changes in the non-tumorigenic HMLE/HMLE-EMT pair (Figure 3, right panels). The “3-carbon” metabolites connecting G3P to pyruvate in the lower chain of the glycolytic–gluconeogenic pathway (i.e., 3-phospho-glycerate, phosphoenolpyruvate, and pyruvate) showed the greatest impact by EMT in the tumorigenic HMLER/HMLER-EMT pair (Figure 3, right panels).

We next constructed two-dimensional PLS-DA models to simultaneously view how the clusters of HMLE, HMLER, HMLE-EMT, and HMLER-EMT cells behaved based on the similarity of their patterns of bioenergetic/anabolic metabolites. Whereas parental HMLE and HMLER cells grouped separately but closely, the EMT process was associated with changes in metabolite levels in such a manner that data samples from non-tumorigenic HMLE-EMT and tumorigenic HMLER-EMT cells mapped far apart (Figure 4, top panels). We then generated PLS-DA variable loading plots to evaluate how strongly each metabolic trait influenced a principal component. Taking a value >0.4 to indicate strong loading, phosphoenolpyruvate, fumarate, and 3-phosphoglycerate explained the separation between the groups in principal component 1, whereas leucine, 6-phospho-gluconate, and fructose-1,6-bisphosphate explained the separation between the groups in principal component 2 (Figure 4, top panels).

To further explore the metabolites discriminating between epithelial and EMT states, the standardized metabolite concentrations were represented in a heatmap for unsupervised clustering. Changes in catabolic and anabolic metabolites segregated epithelial cells from EMT-activated cells irrespective of their background (Figure 4, bottom panels). Indeed, epithelial-to-EMT sample variation was clearly discernible, and the grouped metabolic differences (upper and bottom clusters) were also visible. To evaluate how much accuracy the model loses by excluding each metabolite, we constructed a mean decrease accuracy plot to infer which metabolites were more important for the successful epithelial versus EMT classification. Results showed that 3-hydroxybutyrate, fructose 6-phosphate, fructose 1,6-bisphosphate, phosphoenolpyruvate, and 6-phosphogluconate exhibited the highest values of mean decrease accuracy (or mean decrease Gini score), and therefore were of greater importance in the epithelial *versus* EMT model (Figure 4, bottom panels). Using a metabolite–metabolite Pearson correlation approach, we noted the occurrence of two distinct EMT-related clusters formed from the pool of quantified metabolites in the heat map (Appendix A, left panel).

### 3.2. One Carbon (1C) Metabolites Informing SAM/SAH (re)Methylation and/or Transsulfuration Activities Are Differentially Affected by EMT in Non-Tumorigenic and Tumorigenic Backgrounds

We applied a second inhouse targeted metabolomics platform using ultra-high pressure liquid chromatography coupled to an electrospray ionization source and a triple-quadrupole mass spectrometer (UHPLC-ESI-QqQ-MS/MS) [55,56,57] to quantitatively examine how the folate/methionine bicyclic 1C metabolome might be altered by EMT in non-tumorigenic and tumorigenic breast epithelial cells.

A quantitative, comparative assessment of 1C metabolite concentrations in HMLE-EMT and HMLER-EMT cells and in HMLE and HMLER parental counterparts is shown in Appendix A, and a schematic view of the mean fold changes in 1C metabolite concentrations is presented in Figure 5 (left panels). Analysis of significant 1C metabolic changes occurring post-EMT revealed that the EMT process caused a significant build-up of methionine and homocysteine, accompanied by decreases in 5-adenosyl-methionine (SAM), 5-adenosyl-homocysteine (SAH), and cystathionine in HMLE-EMT cells when compared with HMLE control cells. Conversely, a significant elevation of SAM, SAH, and cystathionine, was found in HMLER-EMT cells when compared with HMLER control cells (Appendix A; Figure 5, left panels). Sample clustering patterns provided by PLS-DA showed a clear separation of both HMLE-EMT and HMLER-EMT cells from HMLE and HMLER controls (Figure 5, middle panels). Considering VIP scores ≥ 1.5 in the PLD-DA model to maximize the difference in 1C metabolic profiles between post-EMT and epithelial parental cells, SAM and homocysteine showed the most relevant post-EMT alterations in the non-tumorigenic HMLE/HMLE-EMT pair, and SAH, cysteine, SAM, and NADH were the subset of 1C metabolites most strongly affected by EMT in the tumorigenic HMLER/HMLER-EMT pair (Figure 5, right panels).

Two-dimensional PLS-DA models simultaneously informing about the behavior of HMLE, HMLER, HMLE-EMT, and HMLER-EMT clusters based on the similarity of their 1C metabolite patterns revealed that the EMT process associated with changes in 1C metabolite levels in such a manner that data samples from non-tumorigenic HMLE-EMT and tumorigenic HMLER-EMT cells apparently mapped far apart (Figure 6, top panels). Loading plots >0.4 to weigh how strongly each 1C metabolic trait influenced a principal component revealed that homocysteine and serine largely explained the separation between the groups in principal component 1, whereas SAH, NADH, and cysteine explained the separation between the groups in principal component 2 (Figure 6, top panels). Unsupervised hierarchical clustering analysis revealed that variations in 1C metabolites segregated epithelial versus EMT cells irrespective of their non-tumorigenic/tumorigenic background (Figure 6, bottom panels). Similar to the bioenergetic/anabolic metabolites, epithelial-to-EMT sample variation was clearly discernible in terms of 1C metabolites, and the grouped metabolic differences (upper and bottom clusters) were also discernible. SAH, cystathionine, and cysteine exhibited the highest values of mean decrease accuracy (or mean decrease Gini score) and, therefore, were of greater importance in the epithelial versus EMT model (Figure 6, bottom panels). Two distinct EMT-related clusters formed from the pool of quantified 1C metabolites in the heat map were observed when using a metabolite–metabolite Pearson correlation approach (Appendix A, right panel).

### 3.3. Mitochondrial Functioning in the Breast Cancer EMT Program Involves Changes in the Utilization of Pathway-Specific Substrates

We next employed MitoPlate™ technology, a novel phenotypic metabolic array, to measure the rates of production of NADH and FADH_2_ from 31 potential mitochondrial energy substrates (https://www.biolog.com/products-portfolio-overview/mitochondrial-function-assays/ 1 December 2022) (Figure 7). 

By using saponin-permeabilized cells and a redox dye added to 96-well microplates containing triplicate samples of a panel of substrates (Figure 7A), we assayed the mitochondrial function of HMLE/HMLE-EMT and HMLER/HMLER-EMT pairs by measuring the rates of dye reduction from electrons flowing through the ETC from substrates whose oxidation produces NADH (e.g., pyruvate, L-malate, α-ketoglutarate, D-isocitrate, L-glutamate, D-β-hydroxy-butyrate) or FADH_2_ (e.g., succinate, α-glycerol-3-P). Cytoplasmic substrates included glucose, glycogen, glucose-1-P, glucose-6-P, gluconate-6-P, glycerol-P, and lactic acid; TCA cycle substrates included pyruvic acid, citric acid, isocitric acid, aconitic acid, α-ketoglutaric acid, β-hydroxybutyric acid, glutamic acid, glutamine, alanine–glutamine, serine, ornithine, tryptamine, and malic acid; and other mitochondrial substrates included acetyl–carnitine + malic acid, octanoyl–carnitine + malic acid, palmitoyl–carnitine + malic acid, pyruvic acid + malic acid, amino-butyric acid + malic acid, ketoisocaproic acid + malic acid, leucine + malic acid.

The electrons donated to complex I or complex II travel to the distal end of the ETC where a tetrazolium redox dye acts as a terminal electron acceptor and changes from colorless to a purple formazan upon reduction. Thus, each of the 96 assays concurrently run in the MitoPlate S-1™ provides different information as substrates follow different metabolic routes using various transporters to enter the mitochondria, and different dehydrogenases to produce NADH or FADH_2_. Heatmap analysis of the metabolic substrate consumption in HMLE/HMLE-EMT and HMLER/HMLER-EMT pairs is presented in Figure 7B. Results revealed a significant augmentation in the utilization of TCA cycle substrates such as cis-aconitic acid, fumaric acid, and succinic acid, in both non-tumorigenic HMLE-EMT and tumorigenic HMLER-EMT cells. Both HMLE-EMT and HMLER-EMT cells utilized other substrates including lactic acid and malic acid-containing combinations such as malic acid + the fatty acid ester palmitoyl–carnitine and malic acid + pyruvic acid, with the latter particularly utilized by mitochondria of HMLER-EMT cells.

### 3.4. Execution of the EMT Program in a Tumorigenic Background Promotes Resistance to Mitochondrial Complex III Inhibitors

Finally, we explored whether and how EMT execution modified the sensitivity of HMLE/HMLE-EMT and HMLER/HMLER-EMT pairs to mitochondrial-centered poisons. To do this, we used the MitoPlate™ I-1, which can measure the sensitivity of mitochondria to 22 diverse mitochondrial inhibitors that directly or indirectly inhibit the ETC. These included complex I inhibitors (rotenone, pyridaben, phenformin), complex II inhibitors (malonate and carboxin), complex III inhibitors (antimycin A and myxothiazol), uncoupling agents (trifluoromethoxy carbonylcyanide phenylhydrazone (FCCP) and 2,4-dinitrophenol), ionophores (valinomycin and calcium chloride), and other chemicals (gossypol, nordihydroguaiaretic acid, polymyxin B, amitriptyline, meclizine, berberine, alexidine, diclofenac, celastrol, trifluoperazine, and papaverine).

We first omitted the saponification step to convert a mitochondrial function assay into a conventional chemotherapeutic screen, and we measured the effects of the agents (each at four graded concentrations) via assessment of tetrazolium dye-based cell viability 48 h after initiation of drug treatment. After normalization of the optical density at 590 nm (purple color) obtained with each agent to those of the (no-drug) positive-control wells included in the MitoPlate™ I-1 plate, a qualitative overview of the fold change results showed that several mitochondrial agents (e.g., amitriptyline, alexidine, diclofenac, celastrol) indiscriminately reduced viability of all cell types irrespective of the non-EMT/EMT phenotype or non-tumorigenic/tumorigenic background when compared with no-drug controls (Figure 8A). To quantify the occurrence of EMT-related changes in responsiveness to mitochondrial agents, we also calculated a comparison score as the absolute ratio between the EMT and non-EMT parental counterparts (Figure 8B). 

Non-tumorigenic HMLE-EMT cells were slightly more sensitive than HMLE controls to the anti-emetic meclizine, whereas HMLER-EMT cells were (at least 1.5-fold) more sensitive than epithelial HMLER counterparts to the natural phenol gossypol, the phenolic lignan nordihydroguaiaretic acid (NDGA), and the antipsychotic phenothiazine trifluoperazine. Remarkably, HMLER-EMT cells were significantly more resistant than HMLER counterparts to the complex III inhibitors antimycin A and myxothiazol, and to the ionophore valinomycin. HMLER-EMT cells were also partly resistant to the FDA-approved drug papaverine, an inhibitor of mitochondrial complex I that was highly cytotoxic against HMLE, HMLE-EMT, and HMLER cells.

As resistance phenotypes to complex III inhibitors in 48 h lasting cytotoxic assays can be argued to arise from adaptive metabolic reprogramming leading to activation of pro-survival processes, we decided to directly assess mitochondrial function in saponin-permeabilized cells using succinate as substrate (Figure 8C). A metabolic substrate that feeds complex II such as succinate will result in a strong flow of electrons via succinate dehydrogenase, which is expected to be inhibited by complex II and III (antimycin A and myxothiazol) inhibitors but not complex I blockers. A saponin concentration of 30 μg/mL has been shown to efficiently permeabilize cell membrane and abolish glucose metabolism without promoting any significant alteration of the mitochondrial metabolism of substrates such as malate and succinate [67]. Validation of the assay in the presence of succinate confirmed a lack of activity of the complex I inhibitors rotenone and pyridaben regardless of the presence or absence of the EMT phenotype. When compared to HMLER parental cells, HMLER-EMT mitochondria were slightly more resistant to the highest concentrations of the complex II inhibitors malonate and carboxin and highly refractory to complex III blockade by antimycin A and myxothiazol (Figure 8C).

## 4. Discussion

The coupled cell fate decision-making processes of metabolism and EMT can be viewed as a key contributor to cancer therapy resistance, tumor immune evasion, and metastasis [68,69]. Our present analysis reveals significant differences in bioenergetic/1C metabolic signatures, utilization of preferred mitochondrial substrates, and sensitivity to mitochondrial drugs when EMT is executed in non-tumorigenic and tumorigenic backgrounds.

We first explored whether the use of central cytosolic/mitochondrial metabolic nodes differ between human breast epithelial cells engineered to acquire a mesenchymal phenotype in the absence or presence of the H-Ras^V12^ oncogene—a well-recognized driver of the cancer metabolic landscape [70,71]. Targeted analysis revealed that “upper” and “lower” glycolysis appear to work at different rates upon EMT activation in non-tumorigenic and H-Ras^V12^-driven tumorigenic backgrounds. Indeed, 6-carbon molecules in the “upper” chain of the glycolytic–gluconeogenic reaction pathway, which connects glucose to G3P, were those most affected by EMT activation in a non-tumorigenic background. Because the concentration of fructose-1,6-bisphosphate mirrors glycolytic flux [72], our data support a scenario wherein the metabolic reprogramming accompanying EMT in a non-tumorigenic background likely involves changes in upper glycolytic enzymes controlling both glycolytic flux and metabolite levels. Conversely, 3-carbon molecules in the “lower” reaction chain in the same glycolytic pathway—also known as the “trunk pathway” connecting G3P to pyruvate—were the most significantly affected upon EMT activation in a tumorigenic background. Because enzyme steps in lower glycolysis do not control pathway flux but carry a higher flux than any biochemically-possible alternative [73], the massive accumulation of phosphoenolpyruvate in tumorigenic EMT cells might reflect their ability to decouple ATP production from phosphoenolpyruvate-mediated phosphotransfer [74], which can act through feedback mechanisms to inhibit glycolysis, thereby allowing for the high rate of glycolysis to support anabolic metabolism.

Examination of the homocysteine–methionine 1C cycle, a metabolic sensor system controlling methylation-regulated pathological signaling [75,76,77], revealed differential changes upon EMT activation in non-tumorigenic and tumorigenic backgrounds. The demethylation pathway generates the universal methyl group donor SAM and the methylation inhibitor SAH, whereas the remethylation pathway converts homocysteine back into methionine by receiving a methyl group from the folate cycle or from choline/betaine metabolism via methionine synthase or via betaine–homocysteine methyltransferase, respectively. Homocysteine is the sulfur-containing precursor that is ultimately channeled to the transsulfuration pathway via conversion to another sulfur-containing amino acid, cysteine, through cystathionine. Accordingly, the evident accumulation of homocysteine in non-tumorigenic HMLE-EMT cells should result from a decrease in its utilization, from altered remethylation to methionine and/or impaired transsulfuration activity [76,77]. The impairment in remethylation and transsulfuration pathways that accompanies non-tumorigenic EMT was characterized not only by the accumulation of homocysteine, but also by the depletion of SAM, cystathionine, and cysteine. Intriguingly, a completely different functioning of the homocysteine–methionine 1C cycle appears to occur when EMT is activated in a H-Ras^V12^-driven oncogenic background. The observed increase in SAM, SAH, and cystathionine in HLMER-EMT cells strongly suggests an augmented transmethylation activity of SAM, whose elevation is known to increase the catalytic activity of cystathionine-β-synthase—the first and rate-limiting enzyme in the transsulfuration pathway [78]. The differential activation and maintenance of methylogenesis and SAM:homocysteine ratios might underlie dynamic and reversible changes in the DNA methylome of EMT cells, an epigenetically conserved mechanism contributing to cellular transformation, tumoral progression, and therapy resistance [79,80,81]. Because the transsulfuration pathway is a highly plastic emergency response mechanism for maintaining cysteine pools and redox homeostasis in harsh tumor microenvironments, it is tempting to speculate that its functioning may enable tumorigenic EMT cells to circumvent nutrient scarcity and high oxidative stress, and to evade drug-induced cell death [76,77,82]. Nonetheless, we acknowledge that one major limitation of the steady-state metabolomic approach employed here is that it does not allow us to discern whether the observed changes in metabolites are causally associated with changes in production or utilization. Metabolic flux experiments utilizing ^13^C-labeled carbon, glutamine, serine, and methionine tracers will be needed as a next step to elucidate the nature of the observed changes within the glycolytic pathway and the methionine–homocysteine 1C cycle in non-tumorigenic and tumorigenic EMT cells.

We next performed functional mitochondrial phenotyping to test epithelial and EMT-activated cells for their ability to utilize bioenergetic substrates. We observed that lactate could support mitochondrial energy production in non-tumorigenic and tumorigenic EMT cells but not in their epithelial counterparts. For many years, lactate had been seen as a metabolic waste product of glycolytic metabolism; however, an ever-growing body of evidence has revealed novel roles of lactate in the tumor microenvironment, either as a signaling molecule or as a metabolic fuel [83,84,85,86]. In the latter regard, tumorigenic EMT cells were notably capable of utilizing not only lactate as a mitochondrial fuel but also the glycolysis end-product pyruvate when simultaneously provided with an additional exogenous substrate such as malic acid. Indeed, not only were high-energy glycolysis end-products utilized at significantly higher rates in EMT cells than in non-EMT cells, but the same was true for the mitochondrial β-fatty acid oxidation (FAO) substrate palmitoyl–carnitine. Mitochondrial FAO has recently been described as a druggable metabolic “gateway” that is activated for EMT cell-state transitions to occur [87]. Previous findings using intact cells revealed that EMT is accompanied by a metabolic infrastructure that enables the scavenging and catabolization of high-energy nutrients such as pyruvate and lactate and the ketone body (and mitochondrial β-oxidation substrate) β-hydroxybutyrate, supporting mitochondrial energy production [88]. We confirm that the mitochondrial phenotype of the EMT phenomenon might constitute an efficient adaptive strategy through which lactate and pyruvate can effectively substitute for glucose as efficient mitochondrial substrates, thereby providing a bioenergetic advantage in hostile microenvironmental conditions involving nutrient starvation. Although it is well established that tumor cells can take-up and oxidize lactate as a fuel under metabolically stressful circumstances when glucose becomes limited [89,90], there is controversy as to whether lactate must first be converted to glucose via gluconeogenesis. The Biolog MitoPlate S-1™ assay directly measures electron flow into and through the ETC from various mitochondrial substrates that produce NADH and FADH_2_ under conditions of saponin permeabilization. Notably, saponin permeabilizes only the plasma membrane, leaving the intracellular membranes of the mitochondria intact while equilibrating the intracellular spaces with incubation medium [67,91]. Our findings, therefore, support the notion that lactate could be imported and converted into pyruvate inside mitochondria to directly feed the TCA cycle in EMT cells.

Finally, we used a modified version of the MitoPlate I-1™ assay to carry out cytotoxic screenings and test the notion that mitochondria-centered metabolic reprogramming is necessary for the survival of EMT cells. With the exception of a slightly augmented sensitivity to meclizine, an antiemetic that indirectly attenuates mitochondrial respiration by targeting cytosolic phosphoethanolamine metabolism [92,93], EMT activation in a non-tumorigenic background had no phenotypic (cell viability) consequences in terms of altered responsiveness to mitochondrial poisons. By contrast, tumorigenic EMT cells showed enhanced sensitivity to several agents, including gossypol, a naturally occurring mitochondrial aldehyde dehydrogenase inhibitor extracted from a cotton plant [94,95], the phenolic lignan NDGA, which promotes mitochondrial depolarization by targeting glutathione oxidation [96,97,98,99], and the antipsychotic trifluoperazine, an inhibitor of mitochondrial permeability transition [100]. Intriguingly, we found an unanticipated resistance to the mitochondrial complex III inhibitors antimycin A and myxothiazol upon EMT activation in a tumorigenic background. Mitochondrial complex I and II donate electrons to ubiquinone, resulting in the generation of ubiquinol and the regeneration of the NAD^+^ and FAD cofactors, whereas complex III oxidizes ubiquinol back to ubiquinone. This raises the question of how tumorigenic EMT phenomena circumvent the cytotoxic effects of blockade of mitochondrial ubiquinol oxidation imposed by complex III inhibitors. Because mitochondrial thioredoxin reductase (TrxR2) can reduce cytochrome c to promote resistance to complex III inhibition upon both antimycin A and myxothiazol treatment [101], it is possible that the complex III-bypassing function of TrxR2 might play a cytoprotective role in tumorigenic EMT cells. However, it has recently been reported that the essential function of mitochondrial complex III for tumor growth is ubiquinol oxidation and not its ability to proton pump or donate electrons to the downstream electron carrier cytochrome c [102]. In this line, we found that tumorigenic EMT cells show enhanced utilization of succinate, which might be consistent with an enhanced complex II activity and an over-reduction of the ubiquinone pool, thereby driving electrons backwards into complex I in a process known as reverse electron transport (RET). In the latter regard, it should be noted that a significant alteration in the electron flow intensity from succinate to mitochondrial complex II and III was indirectly confirmed in saponin-permeabilized tumorigenic EMT cells, in which the mitochondria functioning exhibited partial resistance to respiratory complex II inhibitors and a highly significant refractoriness to complex III inhibitors. Whether succinate-energized mitochondria enabling an enhanced NADH/NAD^+^ cycling, the activation of RET, and/or changes to the structure and organization of the ETC can explain the augmented resistance to complex III inhibition in tumorigenic EMT cells requires further study [103]. Because altered sensitivity to ionophores perturbing ion homeostasis affects not only mitochondrial functions but also other cellular processes ranging from vacuolar function, through translation to stress responses [104], the acquired resistance of tumorigenic EMT cells to valinomycin is likely pleiotropic and multi-faceted.

## 5. Conclusions

Metabolic rewiring, mitochondria functioning, and EMT cooperate to regulate cell fate in non-cancer scenarios such as induced pluripotency [105]. The close cooperation between EMT induction and an active form of RAS is sufficient to trigger malignant transformation of mammary epithelial cells including the acquisition of stem cell-like traits [106,107]. The interdependence of metabolism and EMT also suggests that repurposed metabolic inhibitors could be considered as novel therapeutic interventions for blocking EMT-associated metabolic traits that can trigger cancer recurrence [108]. Our present description of a distinctive landscape of metabolic traits in non-tumorigenic and tumorigenic EMT suggests not only context-specific supportive bioenergetic roles, but likely also instructive roles in determining and maintaining cell fate decisions. Moreover, our findings strongly opine that yet undefined links between metabolism and EMT could compromise the efficacy of metabolic therapies in breast cancer. Accordingly, mitochondrial phenotypes should be carefully evaluated when aiming to use mitochondria-targeting drugs to target EMT-driven biological aggressiveness and therapeutic resistance in breast cancer.

## Figures and Tables

**Figure 1 cancers-14-06214-f001:**
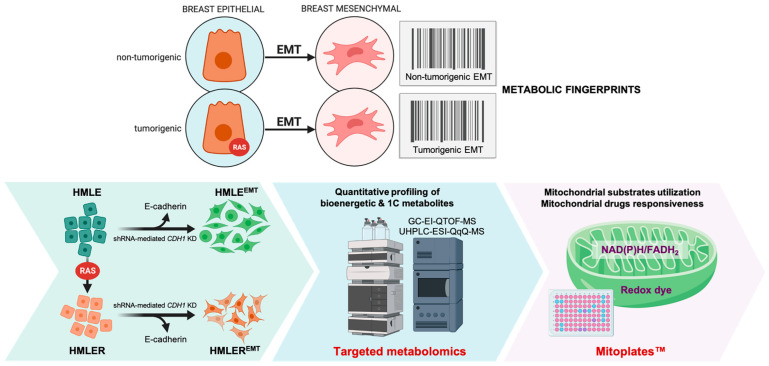
Metabolic fingerprinting of non-tumorigenic and tumorigenic EMT: a metabolomic and mitochondrial phenotyping approach. HMLE/HMLER cells induced for EMT upon loss of E-cadherin and their parental counterparts (HMLE/HMLER) were simultaneously subjected to quantitative screening for bioenergetic (*n* = 30) and 1C (*n* = 14) metabolites and qualitative phenotyping of multiple mitochondria energy substrates (*n* = 31, at low millimolar concentrations (2–5 mmol/L)) and mitochondrial drugs (*n* = 22) using the Mitoplate™ S-1 and I-1 assays, respectively. (shRNA: small hairpin RNA; KD: knock-down).

**Figure 2 cancers-14-06214-f002:**
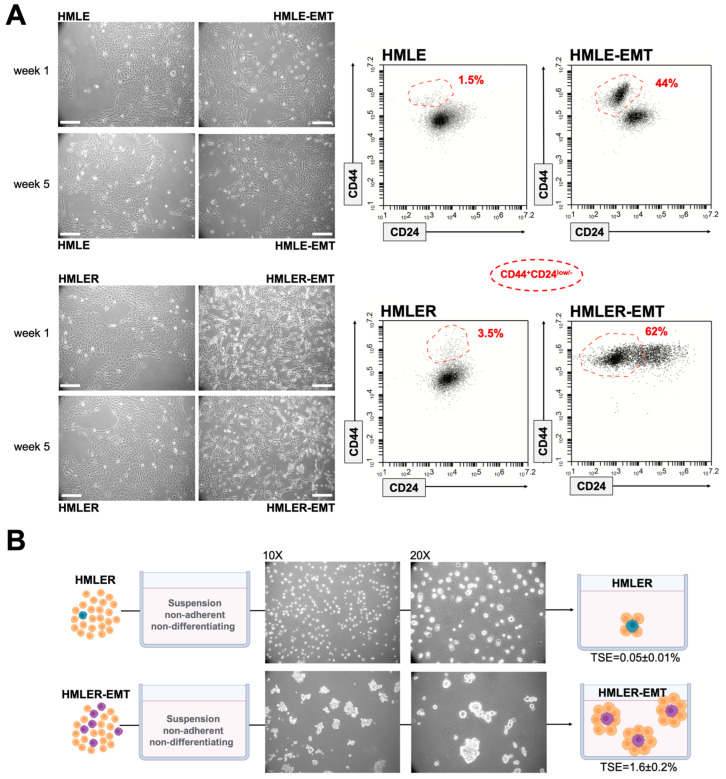
Mesenchymal and tumorigenic traits in HMLE/HMLER cells induced to undergo EMT. (**A**) Left: Representative phase contrast microphotographs of HMLE and HMLER cells modified by shRNA-mediated inhibition of the human *CDH1* gene. Scale bar, 100 μm. Right: Representative flow cytometry plots of CD44 and CD24 expression in HMLE/HMLE-EMT and HMLER/HMLER-EMT pairs (*n* = 3). Red dashed line indicates the distribution of the CD44^+^CD24^−/low^ subpopulation; percentage is indicated for each model. (**B**) Representative light microscope microphotographs of tumorspheres formed by HMLER and HMLER-EMT cells growing in sphere medium for 7 days (10X and 20X magnifications). MSFE was calculated as the number of tumorspheres (diameter > 50 μm) formed in 7 days divided by the original number of cells seeded and expressed as percentage means ± SD (*n* = 5 in triplicate).

**Figure 3 cancers-14-06214-f003:**
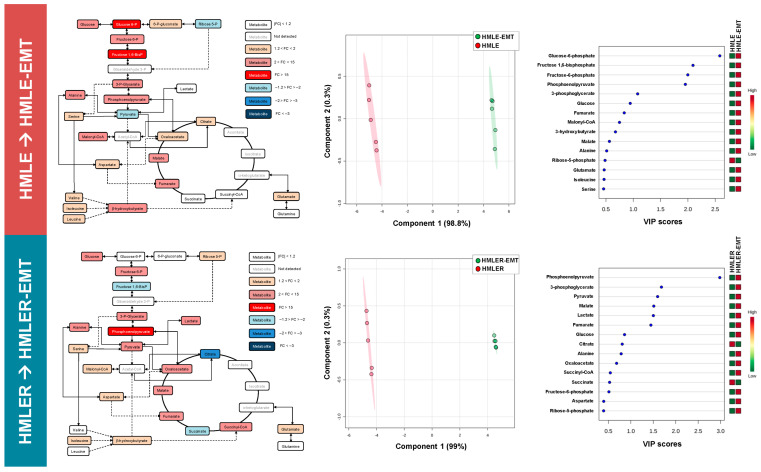
Bioenergetic fingerprints of non-tumorigenic and tumorigenic EMT (I). Left panels: Bioenergetic metabolites from non-tumorigenic HMLE-EMT and tumorigenic HMLER-EMT mesenchymal cells were extracted and quantitatively analyzed by GC-EI-QTOF-MS and compared with those in non-tumorigenic HMLE and tumorigenic HMLER parental counterparts. Significantly increased and decreased metabolites (EMT vs. epithelial controls) are shown using brown-red and light blue-dark blue color scales, respectively. Middle panels: Two-dimensional score plots of the partial least square discriminant analysis (PLS-DA) models of the GC-EI-QTOF-MS-based bioenergetic metabolomic profiling of HMLE/HMLE-EMT and HMLER/HMLER-EMT cells. The X and Y axes represent the combinations of the different bioenergetic metabolites analyzed, showing the maximum separation between groups. Right panels: Key bioenergetic metabolites separating the metabolomic profiles of HMLE/HMLE-EMT and HMLER/HMLER-EMT cells based on variable importance in projection (VIP) in PLS-DA analysis described in the middle panels. The VIP score, which is calculated as a weighted sum of the squared correlations between PLS-DA components and the original variable, summarizes the contribution of the metabolites’ importance in the PLS-DA model. The number of terms in the sum depends on the number of PLS-DA components found to be significant in distinguishing the classes.

**Figure 4 cancers-14-06214-f004:**
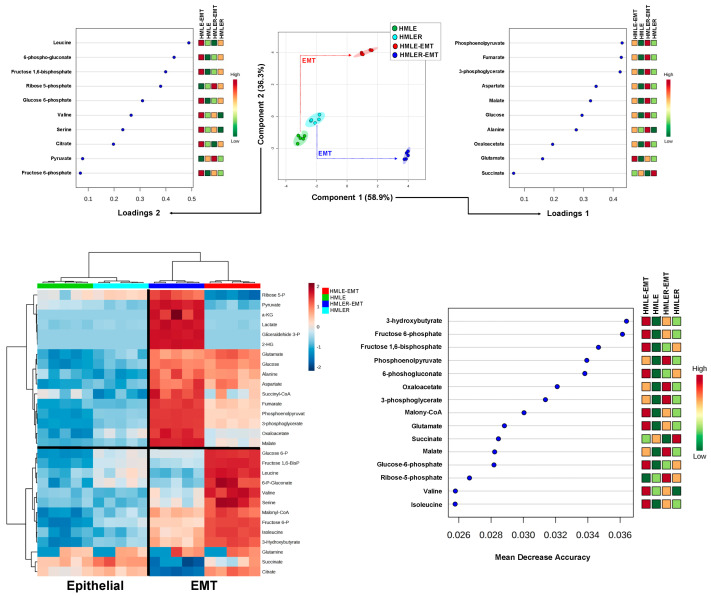
Bioenergetic fingerprints of non-tumorigenic and tumorigenic EMT (II). Top panels: PLS-DA showing four clusters and loading plots based on targeted bioenergetic metabolite profile data derived from HMLE/HMLE-EMT and HMLER/HMLER-EMT cells. Bottom panels: Heatmap visualization, hierarchical analyses, and random forest analysis, of the bioenergetic metabolome of HMLE/HMLE-EMT and HMLER/HMLER-EMT cells. Rows: metabolites; columns: samples; color key indicates metabolite expression value (blue: lowest; red: highest). The list of the first 15 bioenergetic metabolites highlighted by their mean decrease accuracy value is presented for each cell model. Mean decrease accuracy is the measure of the performance of the model without each metabolite. A higher value indicates the importance of such metabolite in predicting each cell line group; removal of that metabolite causes the models to lose accuracy in prediction.

**Figure 5 cancers-14-06214-f005:**
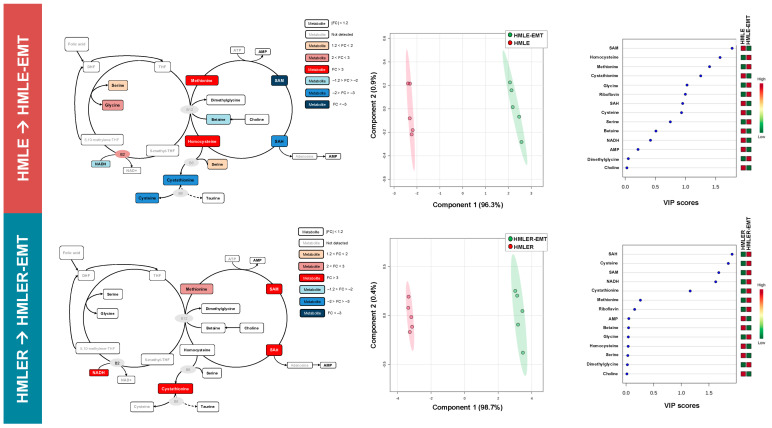
Homocysteine–methionine 1C metabolism fingerprints of non-tumorigenic and tumorigenic EMT (I). Left panels: Homocysteine–methionine 1C metabolites from non-tumorigenic HMLE-EMT and tumorigenic HMLER-EMT mesenchymal cells were extracted and quantitatively analyzed by UHPLC-ESI-QqQ-MS/MS and compared with those of non-tumorigenic HMLE and tumorigenic HMLER parental counterparts. Significantly increased and decreased metabolites (EMT vs. epithelial controls) are shown using brown-red and light blue-dark blue color scales, respectively. Middle panels: Two-dimensional score plots of the partial least square discriminant analysis (PLS-DA) models of the UHPLC-ESI-QqQ-MS/MS-based homocysteine–methionine 1C metabolomic profiling of HMLE/HMLE-EMT and HMLER/HMLER-EMT cells. The X and Y axes represent the combinations of the different 1C metabolites analyzed, showing the maximum separation between groups. Right panels: Key 1C metabolites separating the homocysteine–methionine 1C metabolomic profiles of HMLE/HMLE-EMT and HMLER/HMLER-EMT cells based on variable importance in projection (VIP) in PLS-DA analysis described in the middle panels. The VIP score, which is calculated as a weighted sum of the squared correlations between PLS-DA components and the original variable, summarizes the contribution of the metabolites’ importance in the PLS-DA model. The number of terms in the sum depends on the number of PLS-DA components found to be significant in distinguishing the classes.

**Figure 6 cancers-14-06214-f006:**
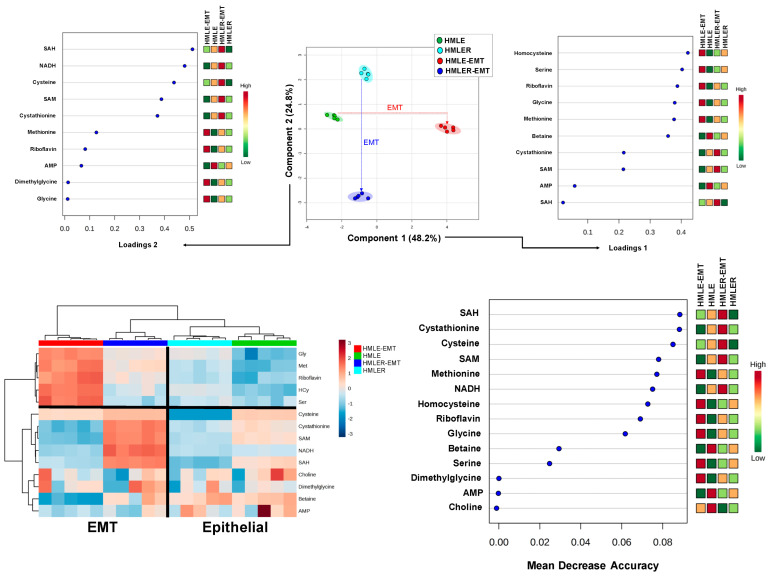
Homocysteine–methionine 1C metabolism fingerprints of non-tumorigenic and tumorigenic EMT (II). Top panels: PLS-DA showing four clusters and loading plots based on targeted homocysteine–methionine 1C metabolite profile data derived from HMLE/HMLE-EMT and HMLER/HMLER-EMT cells. Bottom panels: Heatmap visualization, hierarchical analyses, and random forest analysis of the homocysteine–methionine 1C metabolome of HMLE/HMLE-EMT and HMLER/HMLER-EMT cells. Rows: metabolites; columns: samples; color key indicates metabolite expression value (blue: lowest; red: highest). The list of the first 15 homocysteine–methionine 1C metabolites highlighted by their mean decrease accuracy value is presented for each cell model. Mean decrease accuracy is the measure of the performance of the model without each metabolite. A higher value indicates the importance of such metabolite in predicting each cell line group; removal of that metabolite causes the models to lose accuracy in prediction.

**Figure 7 cancers-14-06214-f007:**
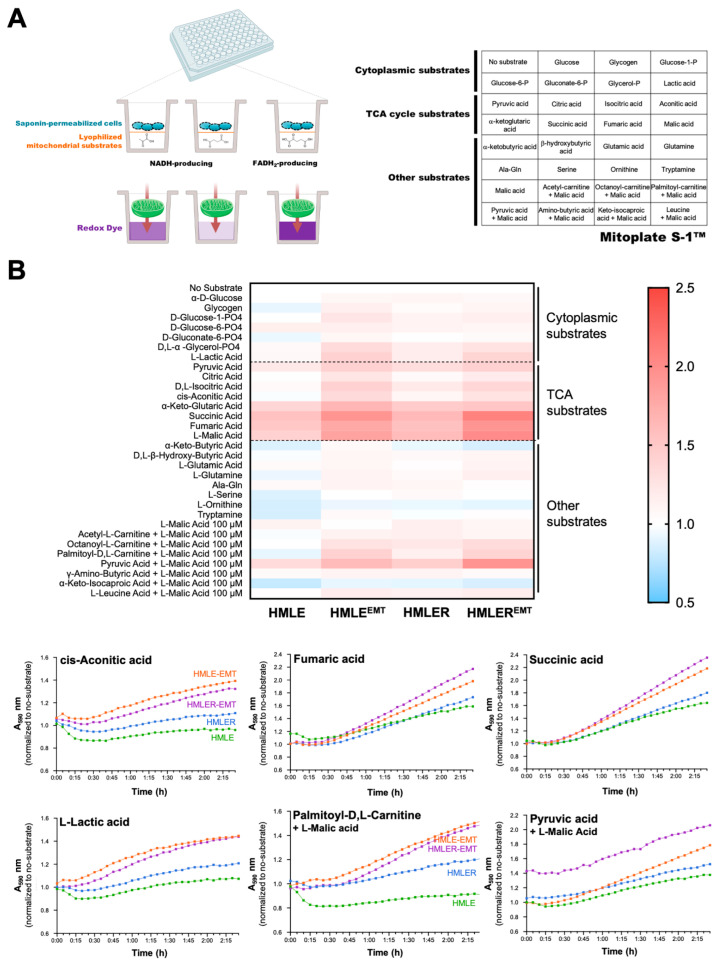
Mitochondrial functioning of non-tumorigenic and tumorigenic EMT. (**A**) Mitochondrial phenotyping of HMLE/HMLE-EMT and HMLER/HMLER-EMT cells using mitochondrial function assays with Biolog MitoPlate S-1 (*n* = 3). (**B**) Top: Representative heatmap of the metabolic substrate consumption (2 h) of fatty acids, glycolysis, amino acids, and TCA cycle intermediates in HMLE/HMLE-EMT and HMLER/HMLER-EMT cells (*n* = 3). Bottom: Representative reduction dynamics of the dye over time measured as absorbance at 590 nm for 2 h at 5-min intervals (*n* = 3).

**Figure 8 cancers-14-06214-f008:**
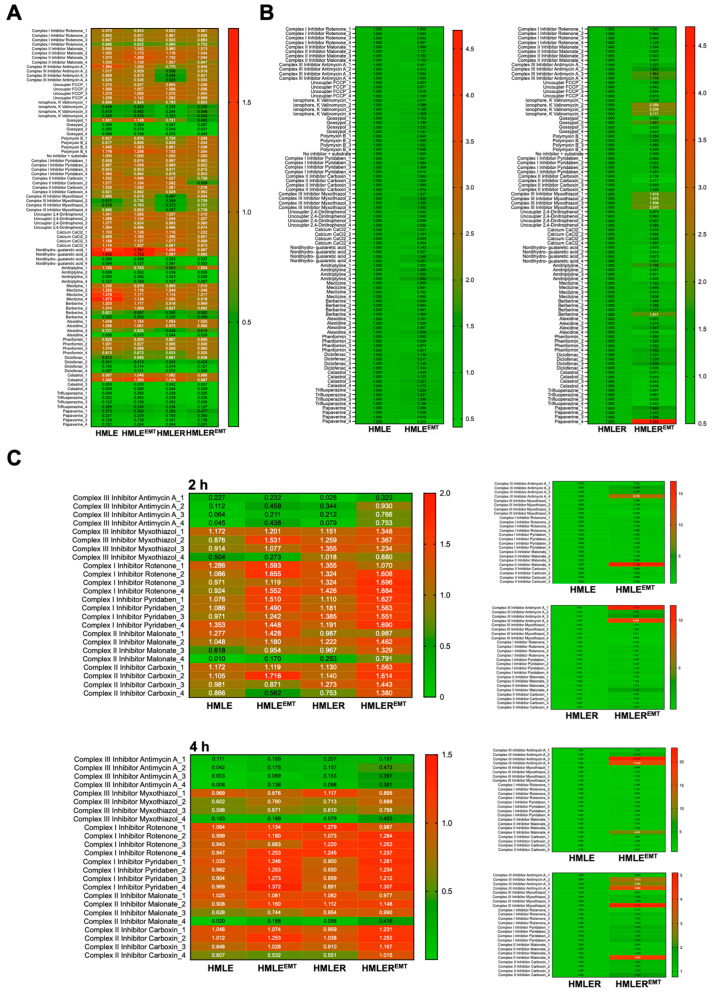
Responsiveness of non-tumorigenic and tumorigenic EMT to mitochondrial-centered drugs. Representative phenetic maps of HMLE/HMLE-EMT and HMLER/HMLER-EMT cells growing in the presence of graded concentrations of mitochondria-centered drugs during 48 h, generated after normalization of the optical density values of each drug concentration at 590 nm (purple color) to those of the positive-control wells included in the MitoPlate™ I-1 plates (**A**) or calculating the absolute ratio between the optical densities at 590 nm of HMLE (=1.0) vs. HMLE-EMT and HMLER (=1.0) vs. HMLER-EMT cells (*n* = 3) (**B**,**C**) Representative phenetic maps of the 2 h (top) and 4 h (bottom) mitochondrial activity in saponin-permeabilized HMLE/HMLE-EMT and HMLER/HMLER-EMT cells in the presence of succinate and complex I, II, and III inhibitors (*n* = 3).

## Data Availability

All data generated or analyzed during this study are included in this published article.

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
