# Peer review of "Metabolomic and Mitochondrial Fingerprinting of the Epithelial-to-Mesenchymal Transition (EMT) in Non-Tumorigenic and Tumorigenic Human Breast Cells"

_cancers, 2022, doi:10.3390/cancers14246214_

Round 1

Reviewer 1 Report

My dears,

Please find my comments in the attached pdf file. Hope you find them useful!

Reviewer 2 Report

In this study, the authors investigated the metabolic fingerprint of cells submitted to EMT in a non-tumorigenic versus tumorigenic context.

Introduction

Line 60-62 : Could the authors develop the part regarding the conflicting literature associating EMT trans-differentiation with tumor metastasis and generation/maintenance of cancer stem cell-like traits.

Line 69 : what do the authors mean by “locking of extreme” ?

Is the KD of CDH enough to induce a full EMT with the acquisition of a migratory and invasive phenotype? Develop the part introducing the cell models and the induction of EMT. (ref 53 and 54).

Results

Line 185-187 figure 2 and Table S1 : Indicate the number of replicates and independent experiments. If the experiment was carried out only once it has to be reproduced.

Idem for the 1C metabolome, indicate the number of replicates and independent experiments. If the experiment was carried out only once it has to be reproduced.

Idem for the Mitochondrial Function Assays with Biolog Mitoplates, indicate the number of replicates and independent experiments. If the experiment was carried out only once it has to be reproduced.

Same comment for the responsiveness to mitochondria centred drugs.

Reviewer 3 Report

E. Cuyas and colleagues profiled some metabolic and mitochondrial functioning differences among four previously established breast cancer cell lines: HMLE, HMLE-EMT, HMLER, and HMLER-EMT. The four cell lines are derived from the same parental line but were genetically manipulated to obtain tumorigenic and EMT transformation. The study reports some interesting results, but some weaknesses/flaws should be addressed to validate the work. 

1. The study relies on the four cell lines that were previously generated and reported from another lab. It is critical and the standard to first validate the cell lines with in-house data. It’s typically unusual, but mishandling/mislabeling of cell line samples can happen during the material transfer process. Furthermore, for genetically manipulated cell lines (in this case, shRNA knockdown), gene expression and cell phenotypes could change as cell passages. Therefore, it is important to first validate that the cell lines still behave the same as reported since this is the very basis of the whole study. Such validation can be achieved following the previous literature. For example, images of cell morphology (non-EMT vs EMT), FACS analysis of the CD44high/CD24low population shift, and growth of tumorsphere. 

2. The authors describe the four cell lines as isogenic pairs. Indeed, the four cell lines are from the same parental reduction mammoplasty tissue. However, the cell lines were not made from the same clone. It is possible that a specific subset of HMLE cells was selected in the HMLER cells to complete the oncogenic transformation. I would like to see at least evidence that the genetic difference between HMLE and HMLER is only the RAS gene. Whole genome sequencing of the two cell lines can be conducted to obtain the data. 

3. The mitochondrial drug phenotyping experiment should be conducted in a more rigorous way. In the current design, cells were treated for merely two hours and an MTT-based assay was used to measure cell viability/drug resistance. The problem is that even though MTT assay is commonly used to measure cell viability, it’s not a good fit for this particular experimental design. First, the MTT assay measures the metabolic activity of cells rather than the number of viable cells (cell death). The less MTT signal only means the cells are less metabolically active, but the cells can still be resistant to drugs. It’s even likely that drug-resistant cells will rewire their metabolic capability to survive the altered environment. Secondly, it’s less likely that two-hour treatment can cause significant cell death. So it’s highly likely that the cells experience acute metabolic change, but are still alive. 

To evaluate the drug resistance, I would recommend doing a longer time treatment such as 5 or 7 days to make sure cell death is the major contributor to the viability. To support the authors’ claims, additional experiments are needed. For example, to confirm that HMLER-EMT cells are more resistant than HMLER to the complex III inhibitors, cells can be cultured both in 2D and 3D and treated with the inhibitors for a few days. Cell numbers/viability or spheroid sizes can be quantified to confirm the drug resistance. 

4. In the method section, more detailed information is needed to describe how the metabolite samples are prepared, such as culture format, cell seeding density, culture time, etc. As it’s known that cells can change their metabolism depending on culture conditions,  these culture details are important criteria for others to successfully reproduce the work. 

5. Quick minor issue. In Fig. 6b, colors (red to blue) are missing in the indicative bar, and there are additional uncropped words above “”Palmitoyl-D,L-Carnitine”. 

Round 2

Reviewer 2 Report

All the points have been addressed by the authors.

Reviewer 3 Report

The authors have adequately addressed my main concerns.